# An Essential Role for Alzheimer’s-Linked Amyloid Beta Oligomers in Neurodevelopment: Transient Expression of Multiple Proteoforms during Retina Histogenesis

**DOI:** 10.3390/ijms23042208

**Published:** 2022-02-17

**Authors:** Samuel C. Bartley, Madison T. Proctor, Hongjie Xia, Evelyn Ho, Dong S. Kang, Kristen Schuster, Maíra A. Bicca, Henrique S. Seckler, Kirsten L. Viola, Steven M. Patrie, Neil L. Kelleher, Fernando G. De Mello, William L. Klein

**Affiliations:** 1Department of Neurobiology, Northwestern University, Evanston, IL 60208, USA; samuelbartley2018@u.northwestern.edu (S.C.B.); madisonproctor2021@u.northwestern.edu (M.T.P.); hongjiexia2019@u.northwestern.edu (H.X.); evelynho2019@u.northwestern.edu (E.H.); a1g5m8@u.northwestern.edu (D.S.K.); kristenhschuster@gmail.com (K.S.); bicca.ma@jhmi.edu (M.A.B.); k-viola@northwestern.edu (K.L.V.); 2Department of Chemistry, Northwestern University, Evanston, IL 60208, USA; hsseckler@gmail.com (H.S.S.); steven.patrie@northwestern.edu (S.M.P.); 3Department of Molecular Biosciences, Northwestern University, Evanston, IL 60208, USA; n-kelleher@northwestern.edu; 4Instituto de Biofísica Carlos Chagas Filho (IBCCF), Federal University of Rio de Janeiro, Rio de Janeiro 21941-902, Brazil; fgmello@biof.ufrj.br; 5Mesulam Center for Cognitive Neurology and Alzheimer’s Disease, Northwestern University, Chicago, IL 60611, USA

**Keywords:** neurodegeneration, neurodevelopment, avian embryo cultures, conformation-sensitive antibodies, tau

## Abstract

Human amyloid beta peptide (Aβ) is a brain catabolite that at nanomolar concentrations can form neurotoxic oligomers (AβOs), which are known to accumulate in Alzheimer’s disease. Because a predisposition to form neurotoxins seems surprising, we have investigated whether circumstances might exist where AβO accumulation may in fact be beneficial. Our investigation focused on the embryonic chick retina, which expresses the same Aβ as humans. Using conformation-selective antibodies, immunoblots, mass spectrometry, and fluorescence microscopy, we discovered that AβOs are indeed present in the developing retina, where multiple proteoforms are expressed in a highly regulated cell-specific manner. The expression of the AβO proteoforms was selectively associated with transiently expressed phosphorylated Tau (pTau) proteoforms that, like AβOs, are linked to Alzheimer’s disease (AD). To test whether the AβOs were functional in development, embryos were cultured ex ovo and then injected intravitreally with either a beta-site APP-cleaving enzyme 1 (BACE-1) inhibitor or an AβO-selective antibody to prematurely lower the levels of AβOs. The consequence was disrupted histogenesis resulting in dysplasia resembling that seen in various retina pathologies. We suggest the hypothesis that embryonic AβOs are a new type of short-lived peptidergic hormone with a role in neural development. Such a role could help explain why a peptide that manifests deleterious gain-of-function activity when it oligomerizes in the aging brain has been evolutionarily conserved.

## 1. Introduction

Soluble oligomers of the Aβ peptide (AβOs) are neurotoxins that build up in brain and cerebrospinal fluid (CSF) of individuals with Alzheimer’s disease (AD) [1,2,3,4]. They also manifest in transgenic AD animal models carrying the human amyloid precursor protein [5,6,7,8,9,10]. Experimentally, AβOs induce memory dysfunction and multiple facets of AD neuropathology, and they are emerging as targets for AD therapeutics and diagnostics (for reviews, see [11,12,13,14,15,16,17,18,19]). Toxicity is a gain-of-function property that develops when oligomers are assembled from monomers [20]. Not all Aβ monomers undergo stable oligomerization; however, the human Aβ peptide comprising 42 amino acids (Aβ42) is unusually prone to do so. Although insoluble amyloid plaques were for many years considered the critical AD toxin, AβOs have become widely regarded as the more disease-relevant form of Aβ [21,22].

Given that human Aβ42 readily generates oligomers that are potent central nervous system (CNS) neurotoxins, it is somewhat surprising that its sequence has been retained through evolution. One possible explanation is that under some circumstances AβOs might serve a beneficial role. This idea is supported by a recent study showing that AβOs can exert anti-viral action, helping protect the brain against herpes simplex virus [23]. A second possible role might lie in neural development, a possibility that is addressed here. Certain changes in the AD brain resemble those occurring in the developing brain, e.g., synapse elimination, selective nerve cell death, and gliosis [24,25,26,27,28]. These changes are pathological in the adult brain but are essential in the developing brain. Synapse loss and nerve cell death in AD models are triggered by AβOs and involve hyperphosphorylated tau [9,29,30,31]; forms of pTau that mediate AβO-induced cell damage have been found to be expressed briefly in the developing brain [32,33,34,35]. If AβOs were to be present during development, they would be expected to be transient, similar to AD-type pTau, as they are absent from healthy adults [2,36].

The current study was carried out to investigate the hypothesis that AβOs are transiently expressed in the developing nervous system and are required for proper histogenesis. Experiments were performed with the developing chick. Chicks are one of a number of species that retain the human Aβ42 sequence [37,38,39,40]), and developing chicks are known to transiently express a phospho-tau proteoform (pSer396/404) stimulated in various experimental models by AβOs [29,41,42,43]. Experiments have focused on the retina, which is widely used as a model in studies of neural development [44,45,46,47,48,49,50,51,52,53,54,55,56]. The highly ordered laminar structure of the retina [57,58,59,60,61] is well-suited for the detection of developmental abnormalities, while the developing eyes of individual embryos are accessible to experimental manipulation.

The results show that AβOs are present physiologically during the development of the embryonic retina. They manifest transiently, and their location and abundance is highly regulated. Experiments with embryo cultures show these AβOs are essential for establishing proper cell placement during retina histogenesis. A requirement for transient AβOs in neural development may be another reason why the human Aβ42 sequence has been preserved in certain species, despite the great potential for neural damage caused by re-emergent AβOs in the aging brain.

## 2. Results

### 2.1. Alzheimer’s-Linked AβOs and pTau Are Expressed in the Embryonic Retina

CNS buildup of amyloid beta oligomers (AβOs) and hyperphosphorylated tau in the adult brain is associated with Alzheimer’s pathogenesis [22,62]. Intriguingly, some proteoforms of pTau linked closely to AD also are expressed in the developing CNS, including pTau recognized by the AT8 antibody in the immature human brain [63] and pTau recognized by the PHF-1 antibody in the immature chick retina [32]. These forms of pTau are stimulated in various AD models by AβOs [9,29,31,41,42,62,64,65,66,67,68]; however, the possible presence of AβOs in the developing CNS has not been investigated.

We therefore asked whether the embryonic chick retina, which expresses PHF-1 tau, might also briefly express AβOs. This possibility is in harmony with the fact that chicks express the same amino acid sequence of Aβ42 as humans [38,69], which favors oligomer formation [20,70,71]. We first tested for the presence of AβOs using a sensitive dot immunoblot assay. This assay was previously developed to detect AβOs in extracts from AD-affected human brains and AD animal models [2,5]. The antibody probe for the assay was the AβO-selective mouse monoclonal NU2 [72]. Soluble extracts were obtained from embryonic retinas at ages when PHF-1 tau was present (embryonic days E7, E14, and E20). The dot blots showed an NU2 signal that became prominent by the second week of development but then decreased (Figure 1). The decrease was somewhat unexpected, as pTau detected by PHF-1 is expressed robustly until after the chicks hatch.

We next tested whether another AD-associated pTau proteoform known to be stimulated by AβOs [29] might also be present in the developing retina. This pTau proteoform is recognized by AT8, a monoclonal antibody that targets human tau phosphorylated at Ser202/Thr205 [73]; this proteoform is different from that recognized by PHF-1, which targets human tau phosphorylated at Ser396/404 [73]. Immunofluorescent imaging was carried out in AT8-labeled sections from retinas at embryonic days 8, 14, and 20 (E8, E14, and E20). As shown in Figure 2, the AT8 epitope was present in the retina, was cell-specific, and was sharply downregulated at the end of development. At E8, the AT8 pTau signal occurred in the ganglion cell layer (GCL) and part of the inner nuclear layer (INL). By E14, its distribution was the most prominent in amacrine cells (ACs), with clearly labeled, well-defined amacrine cells at the edge of the inner nuclear layer (INL). The distribution of AT8 extended into the sub lamina of the inner plexiform layer (IPL) associated with amacrine cells. At E20, the signal for the AT8 proteoform of pTau was virtually gone.

The spatiotemporal expression of the AT8 proteoform was markedly different from the pattern previously seen with PHF-1 [32]. At E20, when AT8 pTau was gone, the PHF-1 proteoform of pTau remained prominently expressed and was not downregulated until the chick was more mature (see Figure 1, ref [32]). An especially notable finding was that the PHF-1 and AT8 signals partitioned to different neuronal populations. PHF-1 is known to be prominent in ganglion cells, whereas AT8 was found to be prominent in amacrine cells. The disparate patterns of the two pTau proteoforms suggests the involvement of cell-specific mechanisms regulating their spatiotemporal expression.

### 2.2. Verification of AβO Identification Using Mass Spectrometry

To substantiate the presence of AβOs indicated by the dot immunoblots, we carried out a molecular level analysis of soluble chicken extracts from E16 retina using mass spectrometry. To begin, we determined that the Aβ peptide from synthetic oligomers showed the expected molecular weight (MW) with a good P-score (not shown), allowing us to continue using this method for the analysis of chicken retina extracts. Proteins in the retina extracts were separated by gel-free electrophoresis (GELFrEE), fractions were assessed by SDS-PAGE, and silver staining was used to validate protein separation (Figure 3A). Putative AβOs were found prominently in fraction 7 at a mass of approximately 45 kDa when probed in a Western blot by the NU2 antibody (Figure 3B). Small amounts of signal were seen at a higher molecular weight in subsequent fractions. Western blots of unfractionated extracts from E14 retina using NU2 also showed the 45 kDa band (Figure 4A). The proteins present in fraction 7 were analyzed by mass spectrometry (Figure 3D–G). We identified the Aβ peptide in this fraction, as demonstrated by a total-ion-signal chromatogram (Figure 3D) and a background-subtracted extracted-ion chromatogram, for which only the signal within the expected *m*/*z* of [M+5H]5+ Aβ peptide isotopic peaks was accounted for (Figure 3E). This was in harmony with the synthetic oligomer analysis. Interestingly, the Aβ peptide proteoform eluted from the chicken retina fractions was found in its oxidized form, as depicted by matched peaks for [M+5H]5+ and [M+4H]4+ Aβ peptide ions (Figure 3F) and by the fragmentation map (Figure 3G), in which fragment ions were matched within 10 ppm mass error to specific positions where backbone bonds were cleaved in the fragmentation process. The results confirmed by mass spectrometry that the GELFrEE fraction showing AβO immunoreactivity at 45 kDa contained the sequence of monomeric Aβ42 (4527 Da). This was confirmed in four independent experiments. The presence of Aβ42 in the fraction comprising mid-sized proteins indicates that the 45 kDa protein was an SDS-stable Aβ 10-mer.

### 2.3. Western Blots Show the Presence of Antibody-Specific AβO Proteoforms

In the GELFrEE fractions shown above, a prominent band at ~45 kDa was recognized by the AβO-selective mouse monoclonal NU2; however, additional minor bands were also seen, suggesting the possible presence of oligomers in the fractions with minor NU2-positive bands. To investigate further, Western blots using unfractionated extracts were carried out with two additional AβO-selective antibodies. One was the humanized monoclonal ACU193, currently in a phase I clinical trial (ClinicalTrials.gov Identifier: NCT04931459), and the other was the well-studied mouse monoclonal NU4 [72,74]. As with NU2, these monoclonal antibodies were generated by immunization with full-sized AβOs and cloned for AβO selectivity. The Western blots of the retina extracts from each of three embryos showed prominent bands detected by all the AβO-selective antibodies (Figure 4), but these bands were detected differentially. Bands were found at ~40 kDa (NU2 and NU4), at ~45 kDa (NU2), at ~72 kDa (ACU193), and at ~104 kDa (NU4 and ACU193). The results indicate that the developing retina expresses at least four prominent sodium dodecyl sulfate (SDS)-stable AβO proteoforms. How the different SDS-stable assemblies relate to each other in detergent-free extracts remains to be investigated.

### 2.4. AβOs Are Expressed in a Cell-Specific Manner

The above results confirm that the E14 retina expresses AβOs that are detectable by AβO-selective monoclonal antibodies (Figure 1 and Figure 3). We next investigated the spatial distribution of the AβOs to determine if their expression in the retina was homogeneous or associated with specific cell types. Immunofluorescence microscopy of NU2-labeled sections revealed a highly differentiated pattern of expression (Figure 5A). AβOs in E14 retina showed a prominent presence in the inner nuclear layer (INL), at the inner edge where amacrine cells localize. An example that confirms AβOs in amacrine cells is shown by the colocalization of AβOs with choline acetyltransferase (ChAT), observed in double-labeled retinas (Figure 5A–C). The overlay shows that virtually all ChAT-positive amacrine were co-stained with NU2 (Pearson correlation coefficient of r = 0.746; [75]). This robust colocalization was seen in the soma and neurites of ChAT-positive amacrine cells. The diffuse labeling in the ganglion cell layer (GCL) may be associated with displaced amacrine cells, as the commercial ChAT antibody used here labeled these cells poorly (Figure 5B). On the other hand, not all AβOs in the amacrine cell region were found within cholinergic neurons, and in the IPL, a sub lamina of AβOs not associated with ChAT could be seen that ran parallel to the AβOs colocalizing with ChAT. The identity of other cell types that express AβOs requires further investigation. The central point here, however, is that AβO expression in the embryonic retina is highly differentiated at the cellular level, a conclusion substantiated by further double-labeling experiments with anti-ChAT and ACU193 (Figure 6). These experiments used a different ChAT antibody (a rabbit polyclonal anti-ChAT serum synthesized by Dr. Michael Mäder and gifted by Dr. Miles Epstein), which in this case robustly labeled the displaced cholinergic amacrine cells in the GCL. As before, virtually all cholinergic amacrine cells expressed AβOs, this time detected with ACU193; however, not all AβO-positive cells were cholinergic. The two double-labeling experiments show that an identifiable neuron population contains AβOs recognizable by two different mAbs, even though they do not detect the same bands on Western blots. Besides helping to substantiate the validity of AβO expression in the developing retina, this result is intriguing in showing that the same neurons appear to express different AβO proteoforms.

### 2.5. AβO Proteoforms Differ in Spatiotemporal Expression

Similarities in the distribution of NU2- and ACU193-targeted AβOs at day E14 led us to investigate their relative expression at other ages. To begin, sections from retinas of embryos at ages E8, E14, and E20 were labeled with NU2. For consistency, all images were taken from the central retina, which matures the most quickly. The results showed that the spatiotemporal expression of AβOs detected by NU2 was highly regulated (Figure 7). An especially salient feature was the pronounced downregulation seen at E20. Over the full time course, the NU2 signal first appeared in the inner retina, spread outward in a cell-selective fashion, and then virtually disappeared as the retina matured. The presence of NU2-positive AβOs in the outer retina can be seen more readily in this developmental study than in the restricted image shown in Figure 4. Control images obtained with secondary but not primary antibodies were dark for this series as well as those below. The period during which the AβOs manifested most abundantly is associated with extensive morphogenic activity [27,76]. As shown later, the presence of AβOs appears to be required for normal morphogenesis to occur.

The AβOs detected by ACU193, unlike those detected by NU2, did not show downregulation (Figure 8). ACU193-positive cells first appeared at the inner surface of the retina, with staining at E8 in the nerve fiber layer (NFL), GCL, IPL, and inner INL. At E14, ACU193 oligomers were particularly abundant in amacrine cells. At E20, they still spanned the retina, appearing in ganglion cells (GCs), in amacrine cells (ACs), in apparent bipolar cells (BCs), in horizontal cells (HCs), and in the retina pigmented epithelium (RPE). It is notable that both ACU193 and NU2 detected AβOs in E14 amacrine cells (Figure 5 and Figure 6), but at E20 only ACU193 AβOs were evident. The data are in harmony with the presence of at least two different proteoforms in the same E14 amacrine cells.

The expression of NU4 oligomers was also investigated and found to be different from both ACU193 and NU2 oligomers (Figure 9). NU4 AβOs at E8 were localized to the nascent NFL, with little presence in the GCL or other cell layers. By E14, NU4 AβOs were evident in multiple layers across the retina but were most prominent in the GCL. In contrast, NU2 and ACU193 oligomers at this age were most prominent in the amacrine cell layer. Within the IPL, the NU4 AβOs were evident within two sub laminae that did not align with the NU2 AβOs in the IPL. At E20, NU4 AβOs underwent major downregulation, similar to NU2 oligomers but unlike the ACU193 oligomers. For direct comparison, the differential patterns of expression of the three types of AβOs are shown side-by-side in Appendix A.

The NU4 proteoform also was studied using retina cell cultures and individual cells isolated from papain-treated retinas. The cultures showed the cell-selective expression of the NU4 proteoform; the inspection of these cells showed an apparent association of AβOs with cytoplasmic vesicles (Appendix A). Selective expression was also evident in acutely isolated cells obtained from E14 retina treated with papain. Some neurons were double-labeled by NU4 and Brn3a, a ganglion cell marker (Appendix A, top) and some were positive for NU4 but not Brn3a (Appendix A, bottom). As in culture, AβOs were evident in the neurites of isolated neurons and showed a punctate, vesicle-like distribution. The possibility that AβOs from such vesicles might be releasable and interact with other cells is consistent with results that soluble retina extracts contained AβOs that bound to cultured retina neurons (not shown).

The formation of AβOs depends on Aβ released by BACE, also known as beta secretase, from amyloid precursor protein (APP), both of which are known to be present in the chick retina [38,69]. BACE and APP development was tracked here using A4 and D10E5 antibodies, respectively (Appendix A). The downregulation of NU2 and NU4 oligomers cannot be explained by the simple downregulation of either APP or BACE, as both were present through E20. BACE distribution at E14 overlapped with NU2 and ACU193 oligomers in the amacrine cell region, but not with the abundant NU4 oligomers in the GCL. BACE-1 expression at E20, however, was robust in the GCL, in harmony with the expression of ACU193 oligomers. The regulatory mechanisms of AβO production and trafficking relative to APP and BACE-1 present an intriguing unknown for further investigation.

### 2.6. Dependence of Retina Morphogenesis on AβOs

As a first step to search for the possible functions of AβOs in the developing retina, we began experiments to investigate histogenesis using ex ovo embryo cultures, which have survival rates of 90% [77,78,79,80,81,82]. We found that ex ovo embryos can be virtually indistinguishable from in ovo embryos with respect to morphology and the expression of AβOs and pTau (Figure 10). Cultured in a shell-free environment, ex ovo embryo cultures are readily accessible for manipulation via intravitreous injection.

We began by assessing the histogenic impact of injecting a BACE-1 inhibitor diluted in phosphate-buffered saline (PBS). A single injection was given at E9, and the retina histology was examined six days post-injection. This is a period of major retina growth and development. The injection controls comprised PBS vehicle only. In addition, as only one eye from each embryo was injected, the other eye served as a non-injected control.

The consequence of injecting BACE-1 inhibitor, but not vehicle, was retinal dysplasia. The laminar organization of the retina was disrupted by hill-like protrusions or folds that occurred predominantly along the central retina (Figure 11A). At higher magnifications, a variety of shapes were evident (Figure 11B,C; Figure 12A). Some disruption appeared as pylon-like cell assemblies projecting from the outer retina to the vitreal space, whereas others were rosette-shaped, forming separate layers. The differences in appearance may in part be explicable by the two-dimensional rather than three-dimensional projection of the tissue. Aberrant retina folds similar to those seen here have been reported for a number of pathological conditions [84,85,86,87,88,89,90]. No perturbations were evident in the negative controls injected with PBS vehicle or in non-injected eyes (Figure 11D,E).

Because inhibiting BACE-1 may influence factors other than AβOs [91], we also tested the effect of injecting the AβO-selective monoclonal antibody ACU193 (Figure 12B,C). The consequences were indistinguishable from those seen after the injection of BACE-1 inhibitor. The injections in these experiments were carried out with 9-day-old ex ovo embryos and sampling for histology was performed at 11 days. For the controls, we included eyes injected with non-specific human IgG and non-injected eyes (Figure 12D,E). As with the PBS-injected eyes, these controls showed no effect. Additionally, there was no effect of ACU193 when it was injected into developmentally stunted embryos, which had been held at 4 °C for 24 h after being placed in culture. Overall, all eight control injections showed no development of laminar disruptions (controls: four PBS-injected; two human IgG-injected; two ACU193-injected developmentally stunted embryos), while all four experimental treatment injections induced dysplasia (two BACE-1 inhibitor-injected; two ACU193-injected). Laminar disruptions caused by antibodies as well as BACE-1 inhibitor are consistent with the hypothesis that AβOs play an essential role in the histogenic development of retinal layers.

## 3. Discussion

### 3.1. AβOs, Neurotoxins Linked to AD, May Be Essential for Neurodevelopment

Amyloid beta oligomers (AβOs) are regarded as neurotoxins linked to the onset of Alzheimer’s disease (AD) in the aging nervous system [14,20,22], but new findings described here show that AβOs also are briefly expressed by the immature nervous system and play an essential role in neurodevelopment. Distinct AβO proteoforms, identified by three AβO-selective antibodies, were found in the embryonic retina, where their expression showed a striking pattern of spatiotemporal regulation. The identification of AβOs was confirmed by multiple analyses, including mass spectrometry. AD-related proteoforms of pTau, which are known to be upregulated by AβOs in cell and animal AD models [29,41,73], showed analogous patterns of expression. Treatments to lower AβO levels caused retinal dysplasia, a disruption of the stratified distribution of retina neurons and synapses. These results identify AβOs as a novel, highly regulated factor that participate in retina morphogenesis.

It is clear that monomeric Aβ has a physiological role [92]; however, the current results describe a second circumstance wherein functional AβOs occur in a healthy rather than degenerating CNS. The first documentation of this was during an innate immune response in which AβOs functioned as part of the anti-viral system of the brain, investigated with respect to HSV infection [23]. Here, rather than responding to external cues, the expression of AβOs was physiological, responsive to programs linked to neural development. These two circumstances, wherein the presence of AβOs is functional rather than pathological, would help rationalize why an Aβ sequence that generates oligomers toxic to mature neurons has been preserved in evolution.

The current findings were obtained using the retinas of embryonic chicks, which generate monomeric Aβ42 comprising the same amino acid sequence as human Aβ42 [38,69]. This sequence readily oligomerizes in vitro, even at low concentrations [5,20]. The same self-associating sequence occurs in a number of species, but not in rodents [93], which is why the use of rodents as AD models has depended on the expression of human transgenes [7]. The chick retina is well suited for studies of neural development [51,76], and other species with human-sequence Aβ42 would be predicted to express transient AβOs that function similarly in neural circuit formation.

### 3.2. Multiple Methods Substantiated the Presence of AβOs and Revealed Distinct AβO Proteoforms

Chick retina AβOs were identified by antibodies previously developed to investigate synthetic AβOs and brain AβOs found in transgenic animal AD models and human samples [72,94]. Besides being detected in situ by immunofluorescence microscopy, retina AβOs in the extracts were characterized by mass spectrometry and Western blots. Mass spectrometry established that putative AβOs were in fact assemblies of the Aβ42 peptide. Western blots showed the presence of several SDS-stable AβO proteoforms that were differentially recognized by the NU2, NU4, and ACU193 monoclonal antibodies. NU2 identified bands at ~45 kDa and ~37 kDa, with the 45 kDa band being the most prominent. ACU193 identified a pair of bands at ~100 kDa and a single band at ~75 kDa, with the top ~100 kDa and -75 kDa bands being the most prominent. NU4 identified prominent bands at 100 kDa and just above 37 kDa. AβO proteoforms of ~54 kDa were previously found in human AD [2] and mouse model AD brain samples [6], while ~45 kDa proteoforms in the human AD brain were found to correlate with cholinergic pathology [95]. A possible modular relationship between the SDS-stable proteoforms in the embryonic retina and the structures found in detergent-free retina extracts is under investigation, as is their relationship to AD brain-derived AβOs.

### 3.3. The AβO Proteoforms Targeted by the Three Antibodies Show Striking Spatiotemporal Regulation, with Patterns That Were Clearly Different from Each Other

Proteoforms detected by all three antibodies appeared by the end of the first week of development, but notable differences emerged over the next two weeks. The pattern of expression detected by NU2 and NU4 was comparable to a molecular wave that began in the inner retina, spread outward in a cell-selective manner, and then dissipated. At the time of greatest expression, roughly E12–16, AβOs detected by NU2 and NU4 were prominent in neurons that were aligned on either side of the inner plexiform layer. The NU2 signal was prominent in the soma of amacrine cells, while the NU4 signal was prominent in the soma of ganglion cells. Within the synaptic layers that make up the IPL, NU2 showed one band with prominent AβO labeling, while NU4 showed two bands; the three synaptic bands were not aligned. Several days later, when the retina was nearly mature, the proteoforms recognized by NU2 and NU4 were virtually gone.

The development of ACU193 proteoforms was markedly different. Most notably, when the others had largely disappeared, the proteoforms detected by ACU193 were still present. Interestingly, both ACU193-targeted and NU2-targeted proteoforms were common to virtually all cholinergic amacrine cells. This is intriguing given that NU2 proteoforms in amacrine cells were downregulated, but ACU193 proteoforms were not. Further investigations to correlate AβO proteoforms with the development of particular retina circuits are just beginning. Given the distinct patterns of spatiotemporal expression, particularly the remarkable disappearance of some proteoforms late in development, the elucidation of the underlying regulatory mechanisms will be of considerable interest.

Whether the formation of AβOs occurred where they were detected is not known. In acutely isolated neurons from papain-treated retinas, the AβOs appeared to occur in cytoplasmic vesicles, in both neurites and soma, and the same was seen in cultured retina neurons. The possibility that AβOs might be secreted from their sites of assembly and diffuse to distal sites is consistent with our findings that AβOs in retina extracts could associate with cultured retina neurons (not shown). Current thinking favors the idea that pathological AβOs and pTau in Alzheimer’s disease are spread extracellularly by exosomes [96,97]. Retina exosomes have been isolated [98], including from embryonic avian retina [99], where they are developmentally regulated [45]. Historically, when first discovered in brain preparations, exosomes were referred to as adherons [100]. Interestingly, no amyloid plaques were observed in the immature retina. The observation that AβOs and plaques are not obligately coupled was first shown in studies of the Osaka mutation, although the Aβ itself comprised a modified sequence [101].

### 3.4. Developing Retina and AD-Related pTau

Besides expressing AβO proteoforms, the developing retina was found to manifest an AD-related pTau proteoform recognized by the AT8 antibody, complementing earlier work [32] that identified a pTau proteoform recognized by the PHF-1 antibody. AT8 targets pSer202 in human tau, whereas PHF-1 targets a pSer396/404 epitope [73]. The spatiotemporal expression of the AT8 signal was localized to amacrine cells and disappeared by E20, a pattern notably different from the PHF-1 signal, which is prominent in ganglion cells and still robust at E20 [32]. The disparate expression of the two pTau proteoforms presumably stems from cell-specific regulatory pathways. Because AβOs are known to upregulate pTau phosphorylated at the AT8 and PHF-1 epitopes in multiple AD models [29,31], the tau phosphorylation pathways in developing retina hypothetically could involve different AβO proteoforms. In harmony with this possibility, the pTau pattern detected by AT8 was similar to that for AβOs detected by NU2, whereas the prolonged presence of pTau detected by PHF-1 was similar to that of AβOs detected by ACU193. Such modular signaling pathways would enrich the mechanisms available for the complex spatiotemporal regulation needed to establish retina circuitry.

### 3.5. Investigations into Possible Functions of AβOs in the Embryonic Nervous System Are Just Beginning

As a first step, ex ovo embryo cultures were established to test if injecting agents that lower AβO levels might cause readily observable anomalies in histogenesis. We found that untreated ex ovo embryos appeared normal with respect to morphology, as previously shown by others [78,79,80,81,82], and that their retinas showed the same stratification of AβOs and pTau as observed in ovo. When embryos were given intravitreal injections of a β-secretase inhibitor or the AβO-selective antibody ACU193, retinal dysplasia was evident. Neurons from the outer retina did not undergo proper stratification, but instead generated aberrant folds in the tissue. The ectopic bands of cells appeared cohesive but extended all the way to the ganglion cell layer. Efforts to identify the ectopic cells and the mechanism(s) by which AβOs influence retina histogenesis are underway. The dysplasia observed here is similar to the disruption of laminar organization reported for retinas subjected to various pathological conditions [86,88,89,90,102,103,104].

AβOs in various AD models induce selective synapse elimination and nerve cell death [20,105]. These are major pathologies in AD [106,107,108]; however, synapse elimination and nerve cell death also are important events in neurodevelopment [109,110,111,112]. Interestingly, PiRB, which is a membrane protein involved in developmental synapse pruning [113], has been identified as an AβO receptor [114]. It thus is feasible that AβOs, in addition to helping establish the stratification of retina nerve cell layers, may play additional important roles during neurodevelopment, such as the pruning of excess synapses and neurons.

### 3.6. Important Questions for the Future

Because the transient expression of AβOs during neural development is a new-found phenomenon, many questions remain to be answered, but three are particularly intriguing. First, the mechanisms that mediate the role of AβOs in cell placement and lamination are unknown, but they plausibly may involve tau. Tau is a multi-functional protein known for its impact on microtubules [115]; however, it also can function in cell–cell interactions, mitosis, and migration [116]. Tau is known to mediate at least some of the toxic effects of AβOs in AD models [30,43,64]. While current evidence is consistent with a tau–AβO mechanism in development, precedents exist for additional possibilities. AβOs added to culture models, for example, influence cholinergic and glutamatergic signaling [117,118], and both transmitters can influence cell morphology and phenotype [119,120,121,122,123].

Second, it remains to be seen whether the retina of humans expresses AβOs in various degenerative pathologies. This possibility is consistent with reports linking Aβ to neurodegenerative disorders of the retina, not only AD [124] but also macular degeneration and glaucoma [124,125,126,127]. In addition, experimental type diabetes in rabbits, another species which has the human Aβ42 sequence, has been shown to induce retina AβOs [128].

Third, how the downregulation of AβOs in the embryonic retina is achieved is a question germane to both neurodevelopment and neurodegeneration. The presence of AβOs and AD-related pTau in the embryonic CNS adds substance to early speculations that AD is an aberrant recapitulation of neural development [129]. The wave of AβO expression that moves in a stereotypic manner across the developing retina is not unlike the stereotypic propagation of pathology that occurs in an AD brain [130,131,132]. The relationship between developmentally transient AβOs and pathological AβOs in aging is unknown but is of considerable significance. The discovery of mechanisms that downregulate AβOs at the end of development may provide insight into why AβOs re-emerge in AD and potentially could provide new targets for therapeutic intervention.

## 4. Materials and Methods

### 4.1. Animals

The use of animals in these experiments was in accordance with the guidelines established by the National Institutes of Health (Guide for the Care and Use of Laboratory Animals, 8th edition. National Academies Press (nih.gov)) and Northwestern University. White leghorn chickens (*Gallus gallus domesticus*) were obtained from Sunnyside Hatchery (Beaver Dam, WI, USA). Eggs were incubated in a Brinsea Ova-Easy Advance incubator (Brinsea, Titusville, FL, USA) at 37.5 °C in 60% relative humidity for between 3 and 20 days.

### 4.2. Materials

All chemicals and reagents were purchased from Sigma (Sigma-Aldrich, Milwaukee, WI, USA) unless otherwise noted below.

### 4.3. Dissection, Fixation, Sectioning, and Immunohistofluorescence

The eyes of E8, E14, and E20 embryonic chickens were dissected and fixed in 3.7% formaldehyde in PBS (2 eyes/50 mL) for 24 h at 4 °C, followed by sequential immersion in a solution of 10% then 20% sucrose-PBS for 24 h each at 4 °C to ensure complete fluid exchange and to cryoprotect the tissue in preparation for frozen sectioning. The eyes were then frozen in 20% sucrose-PBS and sectioned at 45 μm using a microtome (Leica SM2010R; Leica Biosystems, Buffalo Grove, IL, USA) followed by submersion in Tris-buffered saline (TBS; pH 7.6). Only sections obtained from the central transverse plane containing the lens were used for immunostaining. The sections were washed three times for 10 min each in TBS followed by permeabilization in TBS containing 0.3% Triton X-100 (TBS-Tx100) three times for 15 min each. Sections were incubated in blocking buffer (TBS-Tx100; 10% normal goat serum (NGS; Thermo Fisher, Waltham, MA, USA)) for 1 h at room temperature followed by primary antibodies in fresh blocking buffer for 16 h at 4 °C. Amyloid beta oligomers were labelled with mouse NU2 or NU4 (3.14 μg/mL) [62], or humanized ACU193 (1 μg/mL; gift from Acumen Pharmaceuticals, Charlottesville, VA, USA). Phosphorylated tau was labelled with mouse monoclonal antibodies, either PHF-1 (1:2000; Invitrogen, Waltham, MA, USA) or AT8 (0.4 μg/mL; Invitrogen, Waltham, MA, USA). Amyloid precursor protein was labelled with mouse 22C11 (2 μg/mL; EMD Millipore, Burlington, MA, USA). Beta-secretase was labelled with rabbit D10E5 (1:500; Cell Signaling Technology, Danvers, MA, USA). Cholinergic amacrine cells were labelled with goat anti-choline acetyltransferase (ChAT; 1:100; #AB144; EMD Millipore, Burlington, MA, USA) or with rabbit polyclonal anti-ChAT serum (synthesized by Dr. Michael Mäder, University of Göttingen, Germany), a gift from Miles Epstein (University of Wisconsin, Madison WI). The sections were washed three times for 15 min each in TBS-Tx100 and secondary antibodies were applied for 16 h at 4 °C in dilute blocking buffer (1:9 blocking buffer:TBS) followed by three washes for 15 min each in TBS. The secondary antibodies used were Alexa Fluor goat anti-mouse, goat anti-rabbit, donkey anti-goat, or goat anti-human (1:2000; Invitrogen, Waltham, MA, USA). Double-labelling with NU4 and PHF-1 was followed this protocol with the primary antibodies applied simultaneously and then the secondary antibodies applied simultaneously. Sections were immersed in Hoechst stain in TBS (1:2000; Invitrogen, Waltham, MA, USA) for 10 min, washed three times for 10 min each in TBS, and then mounted onto slides using ProLong with DAPI (Invitrogen, Waltham, MA, USA). The slides were imaged using a fluorescent widefield microscope (Leica DM6B Widefield Fluorescent Microscope; Leica Microsystems, Buffalo Grove, IL, USA) using 10× and 20× objectives with the appropriate fluorescent filters. The Leica SP5 II confocal microscope was also used occasionally for these studies.

The double-labelling using anti-ChAT and NU2 followed a different protocol. E14 retina sections were blocked using TBS-Tx100 containing 10% normal donkey serum (NDS; Sigma, Milwaukee, WI, USA) rather than 10% NGS for 1 h at room temperature. The sections were then incubated in anti-ChAT and NU2 simultaneously for 16 h in fresh blocking buffer containing NDS at 4 °C and then washed three times for 15 min in TBS-Tx100. The donkey anti-goat secondary antibody was applied for 16 h at 4 °C in dilute blocking buffer followed by three washes for 15 min in TBS. The sections were then incubated in the goat anti-mouse secondary antibody for 16 h at 4 °C in dilute blocking buffer followed by another three washes for 15 min in TBS. The sections then followed the rest of the above protocol using the Hoechst stain.

### 4.4. Immunoblot

Western and dot immunoblots were carried out in a fashion similar to previous studies [72]. Retinas extracted from E7, E14, and E20 embryos were homogenized in 100 μL of Ham’s F12 medium ((+) L-glutamine; (−) phenol; Caisson Labs, Smithfield, UT, USA) containing protease inhibitors (Pierce™ Protease Inhibitor Mini Tablets, EDTA-free; Thermo Fisher, Waltham, MA, USA). Homogenates were centrifuged for 20 min at 14,000 rpm at 4 °C and the supernatant was collected. The total retinal protein in the supernatant was calculated using Pierce™ BCA analysis (Thermo Fisher, Waltham, MA, USA). Retinal protein within the supernatant (20 ug) was run on a Novex 4–20% Tris-glycine gel (Invitrogen, Waltham, MA, USA) for Western blot at 125 V for 90 min. Gel transfer was performed at 100 V for 1 h at 4 °C. The dot blots used a titration of 27 ug, 13.5 ug, 6.75 ug, and 3.375 ug of retinal protein within the supernatant diluted in Ham’s F12 medium. Blots were blocked with TBS containing 0.1% Tween-20 (TBS-T; pH 7.5) and 5% non-fat milk for 1 h at room temperature. Amyloid beta oligomers were labelled with either mouse NU2 (1.5 μg/mL), mouse NU4 (1.5 μg/mL), or human ACU193 (1.5 μg/mL) diluted in blocking buffer for 90 min at room temperature. The blots were then washed three times for 10 min each in TBS-T, followed by incubation in anti-mouse HRP (1:20,000) for NU2 or NU4, or anti-human HRP (1:5000) for ACU193, diluted in TBS-T containing 5% milk for 1 h at room temperature. Blots were then washed three times for 10 min each in TBS-T and rinsed three times with ddH_2_O prior to imaging. SuperSignal West Femto Maximum Sensitivity Substrate (Thermo Fisher, Waltham, MA, USA) was applied to blots for 1 min and photographed using the Azure Sapphire™ Biomolecular Imager.

### 4.5. Gel-Eluted Liquid-Fraction Entrapment Electrophoresis (GELFrEE)

For the validation of the sample preparation methodology, 1 uL of synthetic AβO standards (76 pmol) was diluted in 30 μL of 66% aqueous formic acid for the disruption of Aβ oligomers into monomers. Prior to mass spectrometry, 30 μL of 5% aqueous acetonitrile was added to this disrupted AβO sample. For the preparation of endogenous samples, a similar protocol was followed. First, a 100μL aliquot of each GELFrEE fraction was submitted to protein extraction via methanol/chloroform/water precipitation as described by Wessel et al. [133]. The extracted protein pellet was then suspended in 30 μL of 66% aqueous formic acid and, prior to mass spectrometry, diluted in another 30 μL of 5% aqueous acetonitrile.

### 4.6. Liquid Chromatography-Mass Spectrometry (LC-MS)

Protein samples were subjected to reversed-phase liquid chromatography (RPLC) using an Ultimate 3000 LC system (Thermo Scientific, San Jose, CA, USA). The samples were loaded onto a trap column (20 mm, 150 μm inner diameter, i.d.) packed with PLRP-S resin (Agilent, Santa Clara, CA, USA) for an initial wash. For protein separation, another in-house packed capillary PLRP-S column (200 mm, 75 μm i.d.) was used. Both columns were heated at 35 °C.

For elution gradient optimization, 1μL of disrupted synthetic AβO preparation was loaded onto the RPLC setup and submitted to an acetonitrile/water gradient. The gradient consisted of a ramp of solvent B from 15 to 50% in 30 min., with a total run time of 60 min. including the column wash (at 95% solvent B) and re-equilibration at 95% solvent A. Solvent A consisted of 5% acetonitrile and 0.2% formic acid in water, while solvent B was composed of 5% water and 0.2% formic acid in acetonitrile. For endogenous sample analysis, either 12 μL (for quantitative/wider window analysis) or 6 μL (for narrow-window analysis) of each resuspended protein sample were loaded onto the RPLC setup. An Aβ-monomer-targeted gradient was performed for all endogenous samples as follows: a ramp of solvent B from 18 to 22% for 36 min, shortly followed by another ramp from 35 to 50% for 10 min, for a total run time of 70 min, including column wash and re-equilibration. The outlet of the column was on-line coupled to a nanoelectrospray ionization source, to which a ~2 kV potential was applied for ionizing proteins.

Wider-window and quantitative mass-spectrometry measurements were performed using an Orbitrap Elite Hybrid mass spectrometer (Thermo Scientific, San Jose, CA, USA). The instrument cycled between a full scan (over a 500–2000 *m*/*z* window), a SIM scan (over a 901–911 *m*/*z* window), and an MS2 scan with same isolation window and activation by high-energy collision dissociation (HCD) at a normalized collision energy (NCE) of 25. The applied resolving power was 120,000 (at 200 *m*/*z*), while the automatic gain control (AGC) target was set at 1 × 10^6^ for SIM and full MS. MS/MS scans were recorded at a resolving power of 60,000 (at 200 *m*/*z*). The AGC target for the selected precursor was 1 × 10^6^. All full MS, SIM, and MS/MS scans were obtained by averaging 4 microscans.

For a targeted analysis of Aβ monomers, narrow-window mass-spectrometry measurements were performed using a Q Exactive HF, modified as described in Belov et al. [134] (Thermo Scientific, Bremen, Germany). The instrument cycled between a full scan (over a 500–2000 *m*/*z* window), a selected ion monitoring (SIM) scan (over a 906.4–907.8 *m*/*z* window, to target the 5+ charged state of oxidized Aβ monomer), and an MS2 scan with same isolation window and activation by high-energy collision dissociation (HCD) at a normalized collision energy (NCE) of 25. The applied resolving power was 120,000 (at 200 *m*/*z*), while the automatic gain control (AGC) target was set at 2 × 10^4^ for SIM and 1 × 10^6^ for full MS. MS/MS scans were recorded at a resolving power of 60,000 (at 200 *m*/*z*). The AGC target for the selected precursor was 5 × 10^5^. All full MS, SIM, and MS/MS scans were obtained by averaging 4 microscans.

### 4.7. Label-Free Quantitative Top-down Analysis

For an initial survey of Aβ monomer presence, merged GELFrEE samples from fractions 6–12 of all studied developmental stages were analyzed by LC-MS. To avoid sample contamination by the disrupted synthetic AβO preparation, following the measurement of Aβ monomer retention time and prior to endogenous sample analyses, both columns were discarded and new, freshly-packed columns were set up on the LC system. Furthermore, to avoid sample-to-sample contamination, a “blank” sample, consisting of pure 33% aqueous formic acid and 5% acetonitrile, was run in-between each consecutive sample. For the absolute quantification of the oxidized form of the Aβ monomer, endogenous samples for which LC-MS analyses showed Aβ monomer were re-run in random order, along with Aβ-monomer standards of known concentrations.

### 4.8. Shell-Free (Ex-Ovo) Embryo Culturing

Embryo culturing was based on the protocols developed by Dunn [77]. Eggs were incubated for 70–74 h before culturing. Clear plastic cups (9 oz. solo cups) were filled with sterile ddH_2_O (180 mL; 37 °C) and covered with plastic wrap to be in contact with the water. All contents of the egg were placed onto the plastic wrap. Eggshells were crushed to a powder and sprinkled onto the albumin, CALM, and yolk. Embryos were then placed into an incubator until injection or dissection. The embryos were maintained in a Thermo Scientific Series 8000DH incubator (Thermo Fisher, Waltham, MA, USA) at 37.5 °C in 80% relative humidity.

### 4.9. Intravitreal Injections

Ex-ovo embryos at E9 received a 2 μL injection of either β-Secretase inhibitor IV (500 nM in PBS; EMD Millipore, Burlington, MA, USA) or ACU193 (5 μg) into the vitreous humor of the right eye and were incubated for the number of days specified in the text before the eyes were dissected for immunohistochemistry. Control injections (2 µL of PBS or hIgG) are described in the text.

## 5. Conclusions

Amyloid beta oligomers (AβOs) are toxins found in the aging brain, where they are linked to AD; however, this study using embryonic chick retinas has shown AβOs can be linked to neurodevelopment as well as neurodegeneration. The results show that AβOs are present physiologically in the embryonic chick retina, which expresses the same Aβ as humans and is widely used for developmental studies. AβOs in the developing retina manifest transiently, with the location and abundance of distinct proteoforms being highly regulated and selectively associated with the transiently expressed proteoforms of pTau. Experiments with cultured embryos show these AβOs are essential for proper retina histogenesis. The results indicate that transiently expressed AβOs might constitute a novel type of negative growth factor, and they are in harmony with the hypothesis that neurodegeneration in AD has its roots in neurodevelopment.

## Figures and Tables

**Figure 1 ijms-23-02208-f001:**
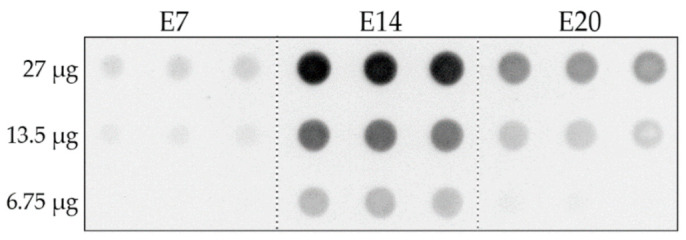
Dot immunoblots detect the presence of transient AβOa in developing retina. Soluble extracts of retinas were obtained from embryonic chicks at ages E7, E14, and E20 and increasing doses were applied to the filter. AβOs were detected using the mouse monoclonal antibody NU2. A major increase was seen between E7 and E14 followed by a prominent decrease between E14 and E20. N = 3 for each embryonic age.

**Figure 2 ijms-23-02208-f002:**
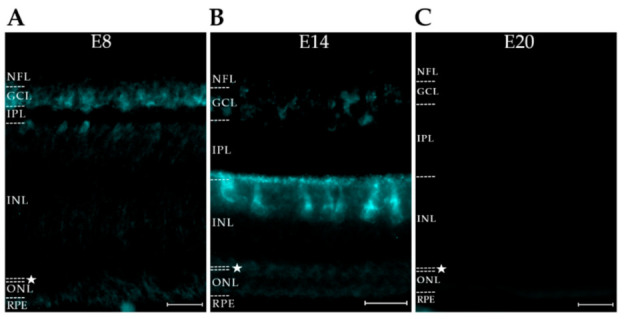
The Alzheimer’s-linked AT8 pTau proteoform is briefly expressed in developing amacrine cells. Retinas from embryonic days E8, E14, and E20 were stained for pTau using AT8 (cyan). (**A**) E8: immunoreactivity is faintly present in the GCL and inner INL. (**B**) E14: immunoreactivity is present at the interface between the IPL and INL with highly specific staining of putative amacrine cells and their processes. Faint immunoreactivity is also present in the GCL. (**C**) E20: immunoreactivity is absent in all layers. The OPL is denoted with a star. Scale bar = 25 μm. N = 2.

**Figure 3 ijms-23-02208-f003:**
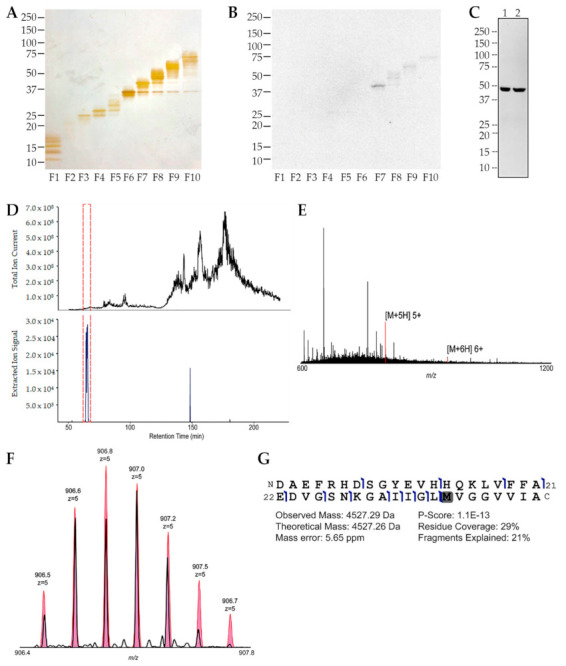
Endogenous AβO expression in the embryonic retina is confirmed by mass spectrometry. (**A**) Soluble extracts of E16 retina were separated into 10 fractions by GELFrEE and each fraction was analyzed by SDS-PAGE and silver staining to establish protein separation. (**B**) Western blot of the GELFrEE fractions showed a prominent signal in fraction 7 at approximately 45 kDa detected by the oligomer-selective NU2 antibody. (**C**) Western blots of unfractionated retina extracts (two separate extracts, 1 and 2, as labeled above each lane) also showed the 45 kDa species. (**D**) Fraction 7 was analyzed by mass spectrometry, and the panel shows the total-ion-signal chromatogram (top) and a background-subtracted extracted-ion chromatogram (bottom). The panel indicates the only signal with the expected *m*/*z* of [M+5H]5+ Aβ peptide. The red dashed box indicates the retention time window in which the oxidized Aβ proteoform was eluted. (**E**) The full intact mass spectrum is shown at the retention time for Aβ elution. Matched peaks for [M+5H]5+ and [M+4H]4+ Aβ are in red. (**F**) The select-ion scan (narrow-window) of the [M+5H]5+ Aβ peptide ion (black) is overlaid on its expected isotopic distribution (pink). (**G**) The fragmentation map is shown for the endogenous Aβ. Fragment ions (blue flags) were matched within 10 ppm mass error to give a specific position where backbone bonds were cleaved. Error in intact mass measurement, the percent of expected fragments observed (coverage) and of fragment peaks matched (explained), and a P-score calculated for confidence in protein identification are indicated. Data confirm that embryonic retina contains an SDS-stable Aβ 10-mer, a result observed in at least four separate experiments.

**Figure 4 ijms-23-02208-f004:**
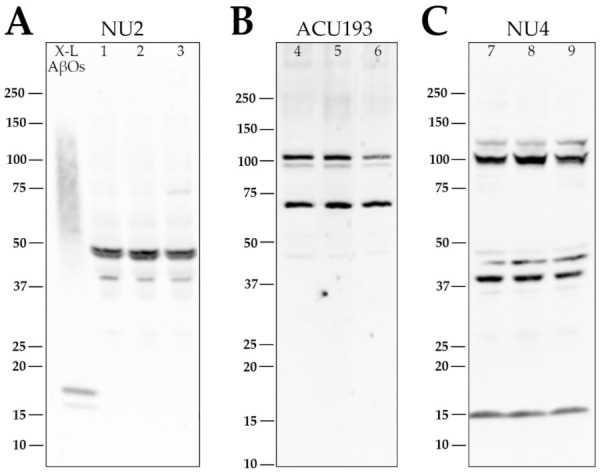
Retina AβOs comprise proteoforms differentially recognized on Western blots by monoclonal antibodies NU2, ACU193, and NU4. Soluble extracts from three embryonic retinas were obtained at E14 and separated using SDS-PAGE (Tris-glycine gel) followed by transfer to a nitrocellulose membrane. This was repeated twice for a total of nine different retinas (1–9) on three separate membranes. AβOs were identified using NU2 (**A**; 1–3), ACU193 (**B**; 4–6), or NU4 (**C**; 7–9). AβOs identified with NU2 were most prominent as a doublet at ~45 kDa with a minor band just above 37 kDa. AβOs identified with ACU193 had prominent bands at ~72 kDa and a doublet at ~100 kDa. AβOs identified with NU4 had prominent bands just above ~37 kDa and ~100 kDa, with fainter bands at ~14 kDa, 45 kDa, and 125 kDa. The ability of NU2, ACU193, and NU4 to distinguish these distinct SDS-stable AβO proteoforms was observed in three separate experiments.

**Figure 5 ijms-23-02208-f005:**
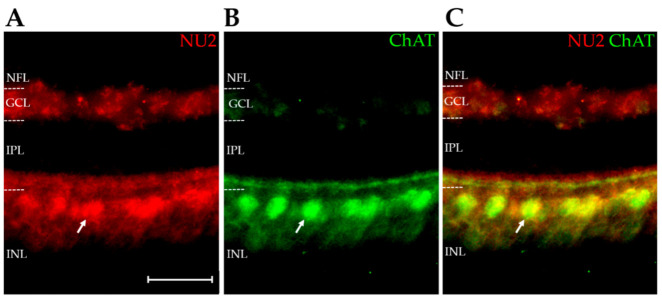
Embryonic cholinergic amacrine cells manifest NU2-targeted AβOs. Retinas from embryonic day E14 were double-labelled for AβOs using NU2 (red) and for ChAT using a commercially available antibody (green). (**A**) AβO immunoreactivity is present in the GCL, outer IPL, and inner INL. (**B**) ChAT immunoreactivity is highly specific to the inner INL and cellular processes projecting into the outer IPL. (**C**) Overlay of NU2 and ChAT staining. Gold represents areas of colocalization. Arrow identifies a cell that is stained by both NU2 and ChAT. Scale bar is 25 μm. N = 2.

**Figure 6 ijms-23-02208-f006:**
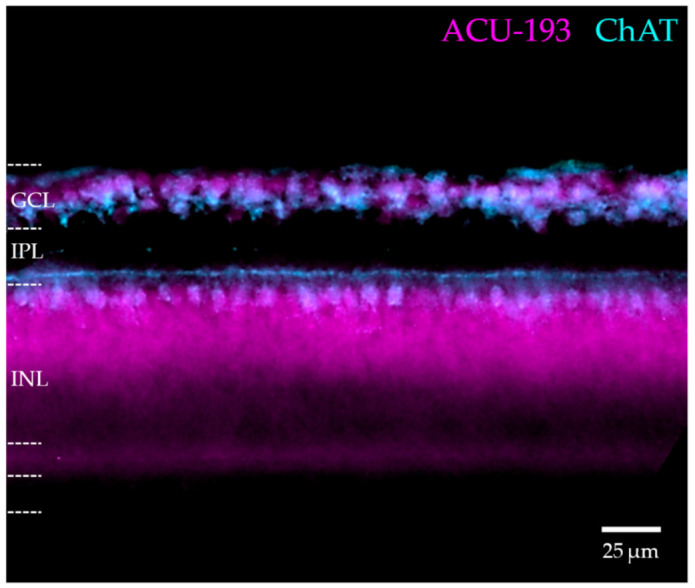
Embryonic cholinergic amacrine cells manifest ACU193-targeted AβOs. Retinas from embryonic day E14 were double labelled for AβOs using ACU193 (magenta) and for ChAT using an antibody that was a gift from Dr. Miles Epstein (cyan). (Magenta) AβO immunoreactivity is prominent in the GCL and inner INL. (Cyan) ChAT immunoreactivity appears in the inner INL, in processes projecting from these cells into the outer IPL, and in putative displaced amacrine cells in the ganglion cell layer. Blue/white represents areas of colocalization. Scale bar is 25 μm. N = 2.

**Figure 7 ijms-23-02208-f007:**
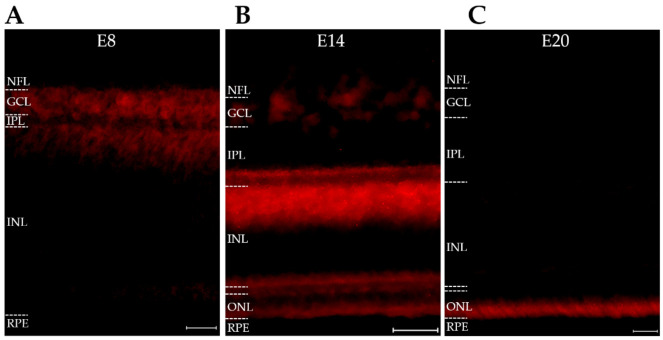
NU2-targeted AβO proteoforms in the developing retina are transient and show complex spatiotemporal regulation. Retinas from embryonic days E8, E14, and E20 were stained for AβOs using the selective mouse monoclonal antibody NU2 (red). (**A**) E8 immunoreactivity is present in the GCL, faintly in the nascent IPL, and the inner INL. (**B**) E14 immunoreactivity is present in the GCL but is otherwise highly localized to the interface between the IPL and INL, as well as flanking the OPL. The outer IPL shows specific sub-banding. (**C**) E20 immunoreactivity is largely gone, localizing only to the outer ONL. As detected by NU2, AβO expression manifests as a wave that progresses from the inner to outer retina and then downregulates. Scale bar is 25 μm. N = 2.

**Figure 8 ijms-23-02208-f008:**
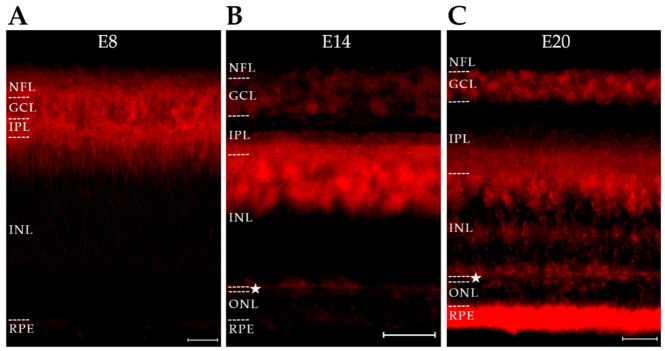
ACU193-targeted proteoforms develop a selective trans-retina expression that is sustained in the E20 embryo. Retinas from embryonic days E8, E14, and E20 were stained for AβOs using ACU193 (red). (**A**) E8 ACU193 immunoreactivity is present in the NFL, GCL, nascent IPL, and faintly in the inner INL. (**B**) E14 ACU193 immunoreactivity is cell specific and faintly present in the GCL with prominent staining at interface between the IPL and INL. There is a specific sub-band in the outer IPL. (**C**) E20 ACU193 immunoreactivity is present through all layers of the retina, prominently in the RPE but also evident in presumptive GCs, ACs, BCs, and HCs. The outer plexiform layer (OPL) is denoted with a star. Scale bar is 25 μm. N = 3.

**Figure 9 ijms-23-02208-f009:**
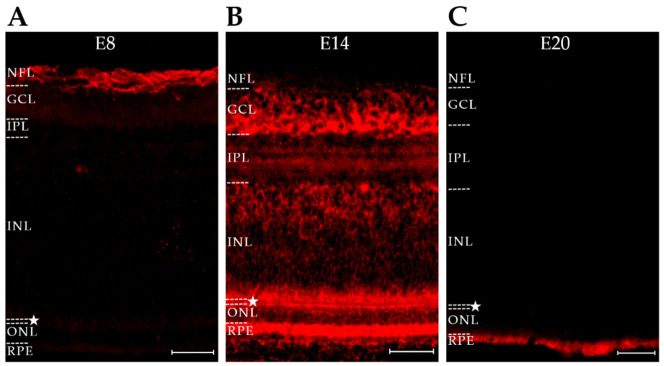
Transient NU4-targeted proteoforms manifest a third distinct pattern of expression. Retinas from embryonic days E8, E14, and E20 were stained for AβOs using the selective mouse monoclonal antibody NU4 (red). (**A**) E8 retina AβO immunoreactivity is localized to the NFL. (**B**) E14 AβO immunoreactivity is spread through all retinal layers, but is most prominent in the GCL, flanking the OPL, and in the RPE; immunoreactivity also is present in two sub-bands of the IPL and the inner and middle INL. (**C**) E20 retina AβO immunoreactivity is gone from most retina layers and localized to the RPE. The OPL is denoted with a star. Scale bar = 25 μm. N = 4.

**Figure 10 ijms-23-02208-f010:**
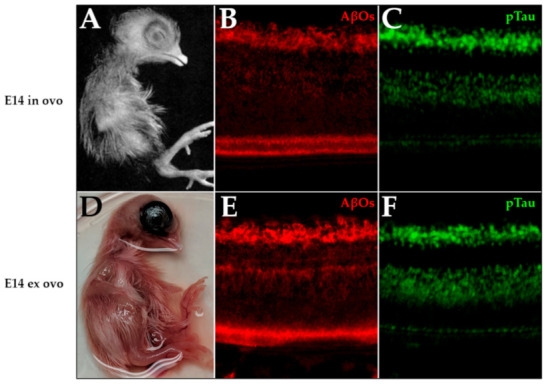
Cultured shell-free embryos resemble embryos maintained in ovo. Embryos were cultured shell-less, as described in the Materials and Methods section, and development was compared with embryos maintained in ovo. (**A**) Embryo incubated in ovo to E14 (from Hamburger Hamilton [83]), Copyright 1951 John Wiley and Sons. (**D**) Embryo incubated in shell-free culture to E14. (**B**) NU4-targeted AβOs (red) and (**C**) PHF-1-targeted pTau (green) in retina of E14 in ovo embryo. (**E**) NU4-targeted AβOs (red) and (**F**) PHF-1-targeted pTau (green) in retina of E14 ex ovo embryo.

**Figure 11 ijms-23-02208-f011:**
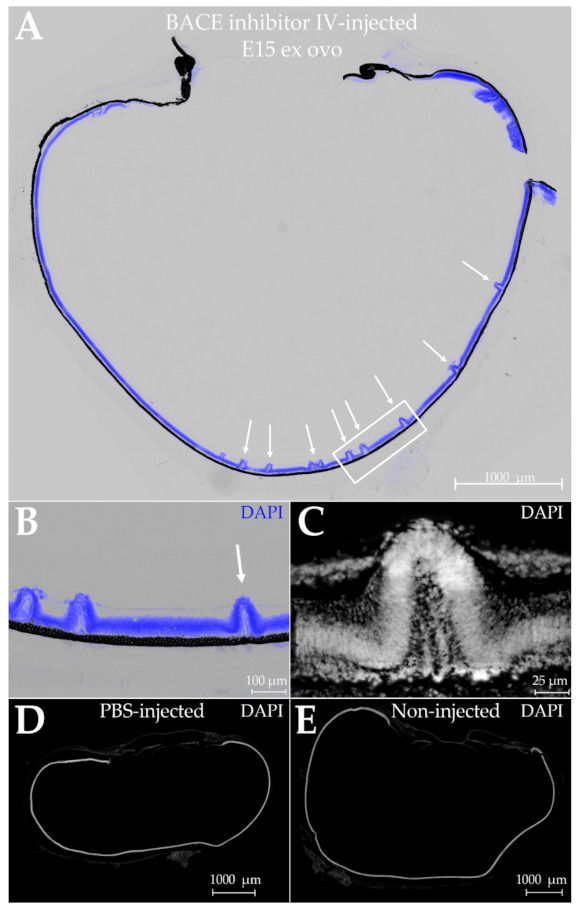
Intravitreal injection of shell-free embryos with a BACE-1 inhibitor induces retinal dysplasia. Ex ovo embryonic eyes were injected with β-Secretase inhibitor IV at embryonic day E9 and dissected 6 days later for inspection of retina morphology. Retinas were stained with DAPI (blue) to identify cell nuclei. (**A**) Laminar disruptions are present at semi-regular intervals. Scale bar is 1000 µm (**B**) Magnified view of selected region (area enclosed in box) shows disruption of all retina layers. Scale bar is 100 μm. (**C**) High magnification view of fold (arrow shown in ‘b’) shows rosette formed within (DAPI in grayscale). Scale bar is 25 μm. (**D**,**E**) No dysplasias were observed in either PBS-injected eyes (**D**) or the non-injected control eyes from each embryo culture (**E**). Scale bars are 1000 µm. N = 2 for each condition.

**Figure 12 ijms-23-02208-f012:**
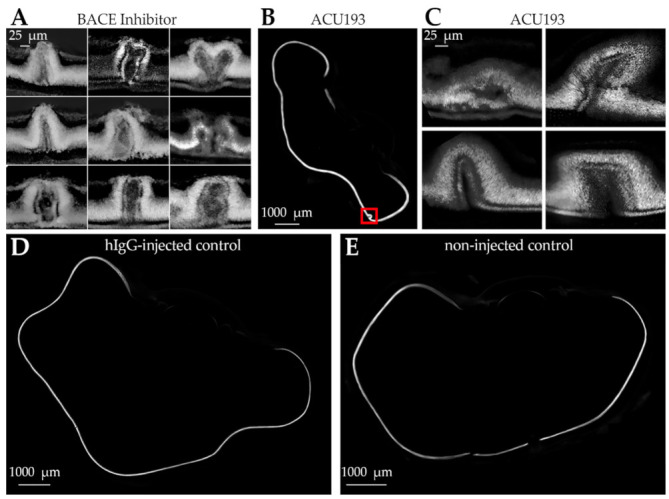
Laminar disruption by injected ACU193 as well as BACE inhibitor indicates a role for embryonic AβOs in retina histogenesis. (**A**) Further examples of dysplasia from ex ovo embryos injected with β-Secretase inhibitor (from embryos shown in Figure 11). Retinas were stained with DAPI (grayscale) to identify cell nuclei. Laminar disruptions of various appearance extend across all retinal layers. Scale bar is 25 µm. (**B**,**C**) Ex ovo embryo eyes were injected with ACU193 at embryonic day 9 and processed for imaging with DAPI at day 11. Dysplasia was evident in the peripheral retina and showed folds similar to those found after BACE inhibitor injections. The red box in (**B**) identifies one such dysplasia. Scale bar in (**B**) is 1000 µm. Scale bar in (**C**) is 25 µm. (**D**,**E**) No disruption was observed in either control eyes injected with non-specific human IgG (**D**) or non-injected eyes (**E**). Scale bars are 1000 μm. N = 2 for each condition.

## Data Availability

Data is contained within the article and Appendix A. Additional details about this study are available upon request from the corresponding author.

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
