# Peer review of "An Essential Role for Alzheimer’s-Linked Amyloid Beta Oligomers in Neurodevelopment: Transient Expression of Multiple Proteoforms during Retina Histogenesis"

_ijms, 2022, doi:10.3390/ijms23042208_

Round 1

Reviewer 1 Report

This research article by Bartley and colleagues, entitled ‘An essential role for Alzheimer’s-linked amyloid beta oligomers in neurodevelopment: transient expression of multiple proteoforms during retina histogenesis’, does an excellent work demonstrating the possible expression of amyloid-beta oligomers (AβOs) as a new type of short-lived hormones with a role in neural development. For this purpose, to test whether AβOs could be functional in development, chicken embryos were cultured ex ovo and then injected intravitreally with either an inhibitor or an AβO-selective antibody to lower AβOs prematurely. The results showed that histogenesis was disrupted, resulting in dysplasia resembling that seen in various retina pathologies. Authors concluded by arguing that embryonic AβOs are a new type of short-lived peptidergic hormone with a role in neural development and may explain why a peptide that manifests deleterious gain-of-function activity when it oligomerizes in the aging brain has been evolutionarily conserved.

The main strength of this original research article is that it addresses an interesting and innovative question, investigating how amyloid-beta oligomers (AβOs), which are normally regarded as neurotoxins linked to the onset of Alzheimer’s disease, could be also briefly expressed by the immature nervous system, playing an essential role in neurodevelopment. In general, I think the idea of this article is really interesting and the authors’ fascinating observations on this timely topic may be of interest to the readers of the International Journal of Molecular Sciences. However, some comments, as well as some crucial evidence that should be included to support the authors’ argumentation, needed to be addressed to improve the quality of the article, its adequacy, its replicability, and thus its readability prior to the publication in the present form. My overall judgment is to publish this article after the authors have carefully considered my suggestions below, in particular reshaping parts of the Introduction and Discussion sections and by adding more evidence.

Please consider the following comments:

  • Abstract: According to the Journal’s guidelines, please present the background, purpose, methods and materials, results, and conclusion proportionally. Also, correct the current total word count, which is 207 words, as it exceeds the Journal’s 200-words maximum.
  • Introduction: The ‘Introduction’ section is well-written and nicely presented, with a good balance of descriptive text about molecular and cell biological aspects of AD, and interpretative illustration of Aβ peptide oligomerization’s role in the pathogenesis of Alzheimer's disease. Nevertheless, I think that more information about pathophysiology and neurological changes of Alzheimer’s disease would provide a better background here. Thus, I suggest the authors to make an effort to provide a brief overview of the pertinent published literature that offers a perspective on structural and functional correlates of age-associated cognitive changes that might indicate neurodegeneration and lead to dementia because as it stands, this information is not highlighted in the text. In this regard, I believe that the statement ‘…AβOs induce memory dysfunction and multiple facets of AD neuropathology, and they are emerging as targets for AD therapeutics and diagnostics’ needs some necessary citations. In particular, according with this sentence, I would recommend citing a recent review that examined pathophysiological basis and biomarkers of AD pathology and investigated molecular signs of neuroinflammation in neurodegenerative diseases, in particular Alzheimer’s disease (https://doi.org/10.3390/ijms21072431). I also recommend a relevant study in which authors investigated age-related impairments in the ability to process contextual information and in the regulation of responses to threat, addressing that structural and physiological alterations in the prefrontal cortex and medial temporal lobe determine cognitive changes in advanced aging, that can eventually cause patterns of cognitive dysfunctions observed in patients with AD/MCI (https://doi.org/10.1038/s41598-018-31000-9). I firmly believe that these improvements will help to provide a more coherent and defined background.
  • Introduction: In according with the previous point raised, when authors stated that ‘Synapse loss and nerve cell death in AD models are triggered by AβOs and involve hyperphosphorylated tau’, I would suggest adding some studies that might address how forms of Aβ and tau protein work together to drive healthy neurons into the diseased state, consistently in the frontal and/or parietal lobes, causing alterations of the frontal lobe that impact memory and error-driven learning in individuals who have a high risk of dementia, may improve the theoretical background of the present article and its argumentation: evidence from an electrophysiological study suggested that medio-frontal ERP signals of prediction error tracks the timing of salient events, and highlighted how alterations in the medial prefrontal cortex could impact on the patients’ capacity of signal errors in the prediction of outcomes (https://doi.org/10.1162/jocn_a_01074). Importantly, in a recent theoretical review that focused on the neurobiology of fear conditioning, the role of the ventromedial prefrontal cortex (vmPFC) was analyzed in the processing of safety-threat information and their relative value, and how this region is fundamental for the evaluation and representation of stimulus-outcome’s value needed to produce sustained physiological responses (https://doi.org/10.1038/s41380-021-01326-4). Finally, authors might also see studies that have focused on this topic (https://doi.org/10.1162/NECO_a_00779; https://doi.org/10.1038/s41386-021-01101-7).
  • Results: Please provide more statistical details to ensure in-depth understanding and replicability of the findings. Specifically, provide more detail about retina’s morphogenesis and tissue engineering after injecting a BACE-1 inhibitor, because it appears unclear how precisely dysplasia occurred as a consequence.
  • Dependence of retina morphogenesis on AβOs: To disentangle whether AβOs may be transiently expressed in the developing nervous system and also required for proper histogenesis, this manuscript focuses on how these AβOs are essential for establishing proper cell placement during retina histogenesis. In this regard, I would suggest, having a more comprehensive and thorough overview on this topic, to also consider quantitation of Aβ oligomerization tendencies in plasma as a potential AD biomarker.
  • Discussion: In this section, authors thoroughly described the results and their argumentation and captured the state of the art well; however, I would have liked to see some views on a way forward. Hence, I ask them to include some thought as well as in-depth considerations, making an effort, trying to explain the theoretical as well as the translational application of their research.
  • Animals: Can the authors provide the specific number of embryonic chickens that were used in the experiments?
  • Also, even though it is not mandatory, I believe that a ‘Conclusion’ section would be useful to adequately convey what the author believe is the take-home message of their study, and therefore provide a synthesis of the data presented in the paper.
  • In according to the previous comment, I believe that the author should make an effort, trying to explain the theoretical implication as well as the translational application of this review, to adequately convey what they believe is the take-home message of their study, and therefore discussing theoretical and methodological avenues in need of refinement, suggesting a path forward in the understanding of cellular senescence in neurons and glial cells’ putative role in AD. In this regard, recent evidence suggests that the application of new methods in Alzheimer’s treatment, such as the Non-invasive brain stimulation techniques (NIBS), have shown promising results in humans (https://doi.org/1097/WCO.0000000000000669). Importantly, I recommend referring to recent studies that revealed that the application of NIBS induces long-lasting effects, noninvasively modulating the cortical excitability, and modulates a variety of cognitive functions: for example, a recent review acknowledged the implementation of NIBS to modulate in general memories (https://doi.org/10.1016/j.neubiorev.2021.04.036). Authors might also consider a review on the efficacy of NIBS and IBS in AD (https://doi.org/10.3389/fpsyt.2018.00201).
  • I would ask the author to insert a ‘Limitations and future directions’ section before the end of the manuscript, in which the authors can describe in detail and report all the technical issues brought to the surface.
  • Figures: I suggest modifying Figure 1 and 3 for clarity and provide higher-quality images because, as it stands, the readers may have difficulty comprehending them. In my opinion, data settings in panel D -E – F are overcrowded and written with a very small font. I suggest to better organizing the graphs’ space in all panels, to provide a better understanding and a direct interpretation of the

Overall, the manuscript contains 12 figures, 3 supplemental figures, and 134 references. I believe that the manuscript may carry important value presenting how AβOs can be linked both to neurodevelopment and neurodegeneration.

I hope that, after these careful revisions, the manuscript can meet the Journal’s high standards for publication. I am available for a new round of revision of this manuscript.

Best regards,

Reviewer

Reviewer 2 Report

The authors provided enough experimental evidence to prove their hypothesis. This is nice research on finding new essential functions of amyloid beta oligomers in neurodevelopment at the embryonic stage. I have the following suggestions to improve the manuscript:

  1. Line 110, please check 'By 14 its'.
  2. Figure 3, all texts are exceedingly small, some are not visible when printed. Please make it bigger and increase all text sizes. Figure 3B shows bands at several different sizes (F7-F10), but figure legend mentioned about 45 kDa only. What about the rest of the band? This image is very small to see clearly. Please rearrange all of the images to provide a better illustration. Figure 3C shows 1   2 without mentioning what. Please mention. Figure 3D-E is very small and not clear. 
  3. Figure 4 doesn't mention the conditions of each well of the gel. Please mention. 
  4. Line 343-345, Injection controls comprised vehicle only (phosphate-buffered saline). Also, as only one eye from each embryo was injected, the other eye served as a non-injected control. Please show in the figure mentioning them as vehicle control, non-injected control, BACE-1 inhibitor etc.
  5. Line 363-364, No dysplasia was observed in un-injected eye from each embryo culture and not observed PBS-injected eyes. Please include the images in the figure.
  6. Line 373-376, Controls injected with non-specific human IgG showed no disruption (n=2). Overall, 4 of 4 eyes injected with AβO-directed treatments showed dysplasia while 8 of 8 eyes injected with control substances showed no dysplasia. Please include the images in the figure for better illustration. Please include images of all used condition for all figures.
  7. Please mention the reason for the use of the materials and method section. For example, why eye samples sequentially emerged with 10%, 20% sucrose solution etc.
  8. The study was done at the embryonic stage. Please mention this information in the first sentence of the conclusion for better clarification.
  9. The present study is especially important, and it may lead to establishing a new era for the amyloid beta study. Therefore, I would like to suggest the authors publish a video article (JoVe or IJMS if possible) for the methodologies that were used in the current study.
